

**Historical Background of Paleo Mega Lake of Rey**
*Hadi Jarahi
*Department of Earth Sciences, Faculty of Sciences, Islamic Azad University, North Tehran Branch, Tehran, Iran.*
**Correspondence**: *Hadijarahi@gmail.com*
**ABSTRACT**
Over the past decade, a vast ancient Lake has been discovered in the central region of Iran,
known as the Paleo Mega Lake of Rey (PAMELA). Considering that the presence of water in
the lake coincides with the existence of human civilizations in the region, it is expected that
references to this lake may be found in ancient Iranian texts. In this paper, we aim to
meticulously examine the provided data while accurately pinpointing the geographical
locations, names, and relevant regions connected to this lake. For this purpose, the majority of
historical texts, travelogues, city histories, royal biographies, and other available sources have
been reviewed. In general, with reference to the mentioned historical texts, it becomes evident
that in many ancient texts, the PAMELA has been mentioned by names like the *Faraxkurt* Lake
and *Saveh* Lake. Numerous ancient sites and structures in the vicinity of the lake have been
identified. Additionally, significant data related to periods when the lake was filled with water
has been acquired. Consequently, there is no longer any room for uncertainty regarding the
presence of the lake during the era when humans resided in the territory of Iran.
*Keywords:* Paleo Mega Lake of Rey, *Faraxkurt* Lake, *Saveh* Lake, Historical text, Travelogue.

**1. Introduction**
In this article, our primary objective is to explore the evolution of writing and calligraphy
in *Iran*. Additionally, we will introduce the largest ancient lake in human history, recently
discovered in *Iran*. This lake presents several intriguing facets and notable challenges, one of
which pertains to its connection with the ancient people of *Iran*. Consequently, we are
dedicated to conducting a comprehensive review of various sources, including books,



travelogues, biographies, and more, to identify individuals who have made references to this
remarkable lake.

## 2. How the Discovery of PAMELA Lake Unfolded

Climate change and the vanishing of ecosystems present significant challenges to the
countries in the Middle East region. Climate studies indicate wet periods in the history of inland
lakes in these areas. The ancient Lake Re, known as the largest lake in human history, spans
across *Iran*, Afghanistan, and Pakistan, and is considered one of the most influential climatic
and ecological factors of the Holocene epoch (Figure 1). Some of the coastal sediments of this
lake were initially brought to attention in *Nazari* studies (Nazari et al., 2010) and Berberian
(Berberian, 2014; Berberian & Yeats, 2016). Ultimately, in *Jarahi* research (H Jarahi, 2021),
it was identified as a unified mega-lake. These findings suggest that our ancestors thrived in a
different climate than what we experience today.
*Maghsoudi* (Maghsoudi, 2021) and *Nazari* (Nazari et al., 2021) have also mentioned the
existence of an ancient lake in the Great Desert region. However, it was in *Jarahi* research
(Habibi, Pourkermani, Ghorashi, Almasian, & Jarahi, 2023; H Jarahi, 2021; H. Jarahi,
Moghimi, Tan, Saygılı, & Karagöz, 2022a, 2022b; Najafian A., Jarahi, & Bayraktutan M.S.,
2022) that the PAMELA theory was first introduced. This theory explores a vast lake, 1.7 times
the size of the *Caspian Sea*, covering the central deserts of *Iran*, parts of *Afghanistan*, and
*Pakistan*. Investigations reveal that this lake began filling up at the onset of the *Holocene*
(*Younger Dryas*) and has existed in these regions for at least over 10,000 years. This timeframe,
considering *Iran's* ancient history, aligns with the rule of various tribes and kings in *Iran*.
Therefore, it is expected that historical texts mention this lake.

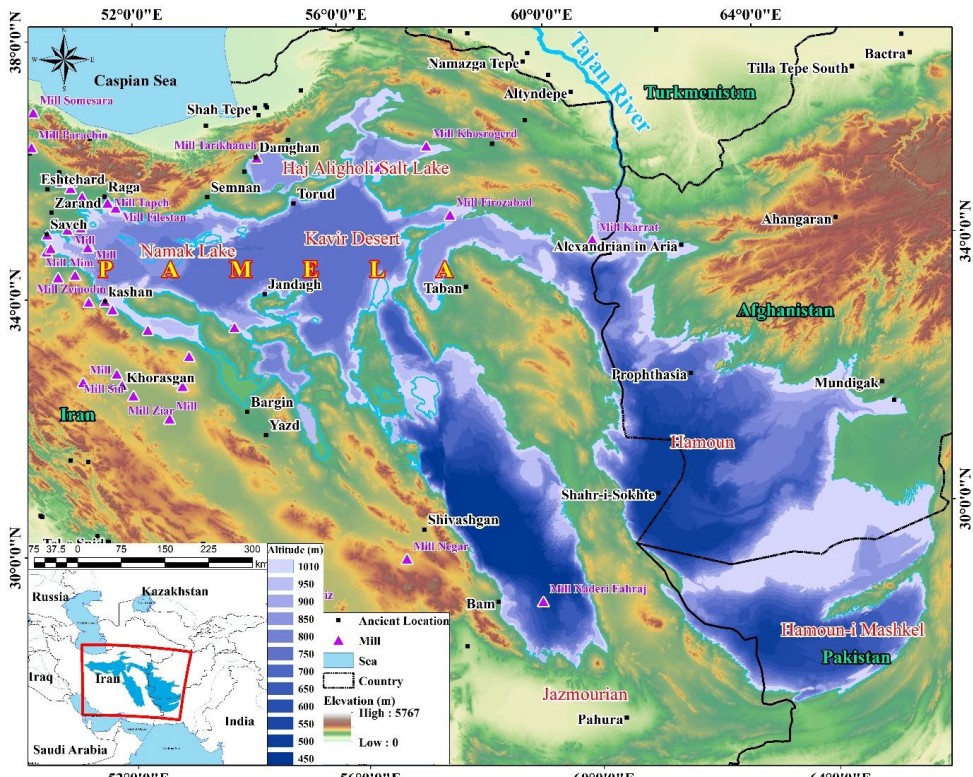

*Figure 1: The geographical location of the ancient Lake of Rey is depicted with changing*

*shades from dark to light blue. This lake covered parts of three countries: Iran, Afghanistan,*

*and Pakistan (H Jarahi, 2021). Important deserts are marked in red, and ancient sites are*

*shown in black. Purple triangles represent mill hills. The positions of the mill hills near the*

*lake's shore correspond entirely to ports and shallow coastlines. The given digital elevation*

*data is accurate to 12.5 meters, obtained from the AleosPalsar satellite.*

## 3. PAMELA in historical text

Paleo Mega Lake of Rey, recognized as one of the most pivotal environmental

determinants in human existence, delineated an epoch characterized by a flourishing maritime

and piscatorial industry. This article seeks to aggregate and scrutinize historical manuscripts

referencing the lake, with the purpose of further scholarly exploration. Consequently, a

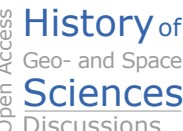

comprehensive selection of over 350 volumes, encompassing literature, travelogues, urban
chronicles, biographies, and more, has been meticulously culled and meticulously examined.
Given their substantial number, specific citation of these sources is omitted in this context.
Subsequently, the ensuing texts have been singled out for in-depth investigation, to be
expounded upon in subsequent sections.
The earliest known literary opus that delves into the subject matter of this lake is the
*Avesta*, dating back to the *Achaemenid* epoch (648 to 330 BCE) (Bleeck & Spiegel, 1999;
Darmesteter & Mills, 2008). Another noteworthy composition, the *Bundahishn*, was authored
during the *Sassanid* era (224-652 CE) (Hale, 2008) and has undergone recurrent revisions
(Oryan, 2021). Within these literary tomes, a myriad of archaic nomenclature and symbolism
is chronicled, some of which have faded into the annals of history. Nevertheless, certain
geographic locales and appellations endure. A salient instance is the expansive water body
known as *Faraxkurt* Lake (*Vouro kasa*). The precise geographical demarcation of this lake
remains a subject of robust scholarly discourse(Green, 2022; Oryan, 2021). *Pourdavoud* has
posited the possibility of it being synonymous with the Caspian Sea (Pourdavoud, 2015), *Bahar*
regards it as the Indian Ocean (Bahar, 2000), while *Derakhshani* identifies it as the Persian
Gulf (Derakhshani, 2003).
The *Avesta* and *Bundahishn* also make allusions to specific geographical and historical
sites contiguous to *Faraxkurt* Lake. As such, it is apparent that *Faraxkurt* Lake was
geographically situated at the base of the Alborz mountain range. Additionally, the *Shushigan*
Mountains (Kerman) and *Khurasgan* (Isfahan) were geographically aligned with the periphery
of *Faraxkurt* Lake (Figure 1). The geographic localization of these *toponyms* underscores their
proximity to PAMELA. Consequently, the discord amongst historians emanates from the lake's
arid state and the paucity of its remnants in the contemporary era.



Curtis (Curtis, 1990) argues that in the expansive arid expanse of the *Great Desert* and
the *Lut Desert*, there once extended a vast lake. *Haghighat* (Haghighat, 1962), recounting the
history of the city of Semnan, reports that some 2,000 years prior to the Common Era, King
*Tahmures* erected the city of *Semnan* on the banks of Lake *Saveh*. He also elucidates the
formation of the *Iranian* Plateau, highlighting that the southern lands of *Semnan* once
comprised coastlines and plains. *Tarih-e-Qomi* (Qomi, 1934) alludes to an extensive lake
spanning from Rey to *Saveh* during the reign of the *Arsacid Kings* (specifically, *Goudarz* in 91
BC). This perspective is further reinforced by the assertions of Strange (Strange, 1930).

*Kateb* (Kateb, 1458), in reference to *Yazdgird II*, one of the Persian monarchs (reigning
from 421 to 439 CE), conveys the following:
*Yazdgird* commanded three generals: *Mibodar*, *Bidar*, and *Eqdar*. He instructed them to
establish three cities. *Mibodar* founded *Mibod*, *Eqdar* established *Eqdā*, renowned for its
association with the *Gabars* village. *Bidar* laid the foundations of *Bidah*. These three cities
were served by a port known as *Bargīn*, located along the shores of Lake *Saveh*. This port was
situated at a distance of 11 *Farsangs* (an ancient *Iranian* unit of length equivalent to
approximately 6 kilometers) from Yazd (Afshar, 1978).

In his travelogue concerning the deserts of *Iran*, *Hedin* (Hedin, 1910) provides a more
comprehensive account of the characteristics of the ancient lake that once existed in this region
compared to other authors. Hedin references ancient *Iranian* texts indicating that during the
reign of *Anushiravan* the Sassanid (531-579 CE), the *Gara Chai* River flowed into the
expansive Lake Saveh. He meticulously traced the remnants of the lake's shorelines to the cities
of *Jandagh* and *Torud* (Figure 1). Hedin also reveals that the city gate of *Jandaq* was



constructed using timber from ships that traversed the Desert Sea, located between *Jandagh*
and *Torud*.
*Zakariya Qazvini*, in "*Athar al-Bilad*" and "*Akhbar al-'Ibad*" (F. H. A. M. Qazvini,
Browne, & Nicholson, 1330; Z. M. Qazvini, 1275), recounts, "In ancient times, there was a
lake near *Saveh* that desiccated and transformed into arable land around the time of the birth of
the Holy Prophet Muhammad (the last Prophet of Islam, 550-570 CE)."
Likewise, *Siroux* (Siroux, 1949) postulates that Lake *Saveh* had desiccated by the time
of the birth of the last Prophet of Islam. *Eghtedari* (Eghtedari, 2022) corroborates *Sirouxs'*
assertions regarding the period of the lake's desiccation. In the book "*Tariqh-e-Qomi*" (Qomi,
1934), based on *Okhravi and Djamal* (Okhravi & Djamali, 2003), there are mentions of Lake
*Saveh* and its desiccation. Additionally, it is reported that Lake *Saveh* was refilled in 1886 CE,
according to a report from *Sadid-o Saltaneh*, an official from the late Qajar period, and this
was reiterated two years later by *Ein al-Dawla* King (Persia, 1888).
Gabriel (Gabriel, 1939) provides invaluable insights into the details of a lake situated in
the current location of the Central Desert (Great Desert). He recounts stories depicting the
desert as an expanse resembling a sea with ships, ports, and lighthouses, among other elements.
Other researchers have also made references to ports known by various names such as
"*Barghin*," "*Barjin*," "*Barajin*," and "*Parchin*" (Pirniya & Afsar, 1991). *Rajabi* identifies the
two cities of *Jandagh* and *Torud* as two forgotten ports in the desert (Rajabi, 2004).

**4. Conclusion**
In general, based on the referenced historical texts, names such as the *Faraxkurt* Lake
and Lake *Saveh* allude to the presence of a large lake in the central region of *Iran*. The
geographical locations of cities and places mentioned in historical texts indicate that the lake's
water level must have been at least approximately 1000 meters higher than sea level. Therefore,





from at least 2000 years BCE until 570 CE, the lake remained filled with water. Ancient cities
like *Saveh*, Rey, *Aveh*, *Kashan*, and others were all situated at elevations ranging from 970 to
100 meters above sea level. Consequently, during this time frame, the lake's water level was
approximately 1000 meters higher than sea level. However, historical records do not provide
information about the extent to which the lake's water level receded after desiccation.
Moreover, in the past two centuries, historical records indicate a re-filling of the lake, at least
in the eastern region (*Salt Lake*).
**Appendix**
**Note about Iranian historical texts**
*Iran* boasts a rich history dating back to ancient times. Discoveries and evidence
unearthed at archaeological sites provide compelling indications of human habitation from the
inception of the Holocene epoch to the contemporary era(Matthews & Nashli, 2022). In the
realm of contemporary geopolitics (Studies, 2020) and across recorded history (Axworthy,
2007), *Iran* has consistently occupied a pivotal and distinctive role. It stands as a vibrant hub
where ideas, ideologies, movements, technologies, and methodologies are conceived,
developed, consumed, imported, and exported in innovative forms, often transcending the
boundaries of Asia and extending into global networks of interaction.
Throughout history, *Iran* has played a central role in the Silk Roads (Frangipane, 2015),
a role that, in a modern context, serves as a crucial bridge connecting China to the
Mediterranean Sea (Griffiths, 2021). The methods of writing in ancient *Iranian* societies have
been widely dispersed since 3000 BCE, and written artifacts have endured across a diverse
array of locales and contexts. Written sources, particularly those inscribed in long-extinct
languages, require specialized skills for their reading, comprehension, and
interpretation(Matthews & Nashli, 2022). For archaeologists, a significant concern pertaining
to these texts lies in the challenge of contextualizing them historically: who were the authors,



who possessed the ability to decipher them, and how were they employed within the field of
archaeology?
Early European explorers and traders, such as *Pietro della Valle* in the early 17th century
and *Cornelius de Bruin* in the early 18th century, have contributed reports and cartography
documenting some of *Iran's* prominent locales and structures, including *Persepolis* (Swanick,
2012; Weerdenburg & Drijvers, 1991). The journey toward unlocking the wealth of ancient
*Iranian* sources commenced with *Georg Grotefend's* identification of *Achaemenid* kings'
names in 1802 through deciphering ancient Persian inscriptions at *Persepolis* (Swanick, 2012),
a breakthrough that substantially enriched our understanding.
An achievement that was significantly increased by *Henry Rawlinson's* decipherment in 1840-
1840 of the trilingual *Bisotun* inscription (*Darius the Great*), near *Kermanshah* (Larsen, 1996;
Peter, 2009; Rawlinson, 1846).
The nascent development of archaeology in *Iran* was intimately connected with the
burgeoning *Iranian* nationalism during the Qajar period, particularly under the rule of *Naser*
*al-Din Shah* (1846-1896)(Abdi, 2001; Goode, 2007). Consequently, the history of writing and
script in *Iran* has been inextricably linked to an evolutionary odyssey. The texts currently at
our disposal represent but a fraction of *Iran's* ancient heritage.
**Acknowledgment**
I am grateful to Dr. Saeed Oryan at Tehran University is appreciated for his valuable advice
and guidance in locating the names of his book.
**Author Contributions:** H.J.: formal analysis, fieldwork, methodology, investigation, and
writing the manuscript.
**Funding:** This research received no external funding.



**Conflicts of Interest:** The author declare that the research was conducted in the absence of
any commercial or financial relationships that could be construed as a potential conflict of
interest.

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

*on the Iranian Monuments*: Nederlands Instituut voor het Nabije Oosten.
