# Peer review of "Historical Background of Paleo Mega Lake of Rey 1 2 \*Hadi Jarahi 3 Department of Earth Sciences, Faculty of Sciences, Islamic Azad University, North Tehran Branch, Tehran, Iran. 4 \*Correspondence: Hadijarahi@gmail.com 5 ABSTRACT Over the past decade, a vast ancient Lake has been discovered in"

_History of Geo- and Space Sciences, 2024_

## Referee Comment (RC1)

I have been conducting Holocene climate history in southern Iran, examining the vegetation and lake histories of several sites in the Zagros Mountains, and have linked it to Metaphysical Climate models based on the climate records of Rafsanjan, Kerman, and Zahedan. We have two publications in review regarding the Holocene climate history of Iran that includes a reconstruction of the precipitation history of southern Iran for the last 13,000 years. My discussion regarding Mega-pluvial Lake Rey will include some reference to that research.

In this paper, Hadi has tried to winnow out the many references to bodies of water in Iran over the past 2,500 years ranging from Achaemenid Empire to as recently as the last few hundred years. The references come primarily from areas of northwestern, north central and eastern Iran. There are a few scattered references as far south as Kerman, but they do not seem to extend into the adjacent, Hamun-e Jaz Murian or Jazmurian Basin just adjacent to the southern margin of Lake Pamela. Mr. Jarahi provide a map to put the lake into context. It is interesting that during the Achaemenid period Alexander the Great's army would have had to wade or swim through the Afghani extension of the lake.

First to the citations that Mr Jarahi provides. They are numerous, they mention locations on the landscape but no descriptions of the lake itself, in most cases. It would be helpful if when he used these citations, he would mention the descriptions of the lake that they contained. The problem with these many citations, is being able to confirm that they all refer to the same lake or possibly to individual lakes scattered around Iran at these times.

Is there physical evidence that a lake that large and extensive has ever existed? Is there climate evidence that indicates that a lake that large could ever have existed. That is what is lacking.  Now where is there any evidence given for a water source, a stream or streams that could have maintained such a lake. Is there any evidence in the pollen records from the lakes in the Zagros Mountains that indicate when such a lake may have existed?  So the climate models indicate when such a lake might have been created by the right mix of rainfall and temperature, so that evaporation rates would have been lower a such a lake might have been maintained. The real problem is that in order to maintain such a lake with a surface area that has been suggest, means that either there must have been incredible water sources, or temperatures had dropped significantly to reduce evaporation from a lake that extensive. Let us look at Tehran.

[Figure]

[Figure]

What is clear is that prior to 10,000 years ago precipitation may have been high enough and temperatures low enough to have supported pluvial lakes. And in fact, sediments from several basins around Iran indicate that there were lakes in those basins at that time. But it is also clear that if we look at the dating of those sediments, they are not in correspondence. The increase in rainfall after 6,000 years ago, is slight (look at the mm numbers on the right hand side of each diagram. The increase is barely reflected in the annual precipitation, and temperature is not going down, so evaporation rates remain high. If you look at spring and summer precipitation there are some brief intervals of precipitation increase with slight decreases in temperature as well. Those were sufficient to result in marsh expansion and the expansion of grasslands around the lake of the Zagros Mountains, i.e., Zeribar, Mirabad, and Maharlou. But they are barely reflected in the sediment records of the playas that have been studied.

If we look at the record from Kerman, at the southern range of pluvial lake Pamela, we have a similar result, just the details are slightly different, but the timing of events remains similar.

[Figure]

[Figure]

As with the Tehran record the early Holocene as the most probably period for a pluvial lake in the small basins of southern Iran. And it is the middle to late Holocene that has several increases in precipitation, whose timing parallels shifts to grassier landscapes and marsh expansion around the lakes in the Zagros Mountians. In particular, is the period around 4,000 years ago, and a couple of times during the last 2,000 years, but note that temperatures are not reduced significantly enough to increase effective rainfall.

So, the question is do many of the accounts that are reported in the literature describe the expansions of smaller, individual lake basins scattered around Iran, rather than describing

one large lake.

I suggest that Mr. Jarahi, groups his citations according to years before the present of the account, and the specific location.

And then he should extract the descriptive details from those citations to determine what is being described.  Lake size, salinity if there is that information, etc.

He might generate a table using those criteria as well. It might make identifying what is being described much easier.

At that point, other investigators could compare the information that Mr. Jarahi has gathered with the scientific record, and we might be able link the descriptions to known scientific lakes. That would create a very rich database of literary and scientific evidence for Iran's past environmental history.

---

## Author Comment (AC1)

Reviewer 1 response:

While thanking the reviewer for his careful opinion, the following points can be mentioned in connection with his comments:

- This article only refers to historical references and tries to present historical documents without prejudice and in an integrated manner. But in the article (Jarahi, 2021) the main topic of the lake and some of its evidence are presented.
- Regarding the climatic evidence, unfortunately, as the reviewer has pointed out, there is no high-resolution study in Central Iran. From Zaribar Lake (in the west) to Jazmurian (in the Southeast), there is a climate information gap. We are working on this issue with the help of two European and American universities, and the related article will be ready in a year. The mentioned article is specially dedicated to climate discussion.
- This article is the result of studying nearly 140 thousand pages of ancient Iranian history texts. From the old books of Avesta and Bondaheshn to contemporary history, they have all been reviewed. In the meantime, there are very limited mentions that have addressed the lake. The only document that talks about the water quality of the lake and its depth is related to one of the Qajar kings named Ain al-Dawlah (Persia, 1888). Other sources are not so accurate.
- As the honorable referee has mentioned, this is a very important issue and requires years of team effort. Over the past 16 years, extensive research has been done across the country to collect these documents. However, in a scientific work, there are always shortcomings that require further studies. As mentioned, we are trying to separate the documentation of PAMELA Lake and examine it with different scientific topics such as climatology, history, geology, archeology, and even seismology. Here only history is considered.
- A table of historical documents has been prepared separately for each period, which is added to the manuscript. According to the opinion of the reviewer, this table can help to understand the main topic of the article.
  Sincerely yours,
  Hadi Jarahi

Table 1 summarizes Iran's ancient historical documents, which refer to Lake PAMELA. This indicates a decrease in water levels from 2000 BC to 1888 AD.

| No. | Period | Document | Lake name | Area/Explanation | Altitude |
|---|---|---|---|---|---|
| 1 | 2000 BC | Haghighat, 1962 | Saveh Lake | Tahmurisdin Shah built the city of Semnan on the edge of Saveh lake | 970 m |
| 2 | 330-648 BC | Avesta | Farakhkort Voerokashs | Alborz mountains in North of Iran, Shoshiagan in Kerman, Khurasgan in Isfahan, are located in the shoreline of Lake. | - |
| 3 | 91 BC | Qomi, 1934 and Strange, 1930 | Saveh Lake | A large lake from Rey to Saveh during the Parthian kingdom | 970 m |
| 4 | 652-224 AC | Bondaheshn | Farakhkort Voerokashs | Same as no.2 | - |
| 5 | 439-457 AC | Kateb, 1458 | Saveh Lake | Meybod, Bideh and Aghda Cities are located in the beach. The Bargin City was their port. | 970 m |
| 6 | 550-570 AC | Qazvini, 1275 | Saveh Lake | Saveh City was a port of Lake Saveh | 970 m |
| 7 | 953-979 AC | Hedin, 1910 Siroux, 1949 Eghtedari, 2022 Qomi, 1934 Okhravi and Djamali, 2003 | Saveh Lake | Jandagh in Isfahan and Torud in Damghan Cities are the port. | 800 m

800 m |
| 8 | 1888 AC | Persia, 1888 | Saveh Lake | Water level rising cased to Huzi Soltan and Namak Lake connect together | 800 m |
| 9 | - | Gabriel, 1939 Pirniya and Afsar, 1991 Rajabi, 2004 | Saveh Lake | Lighthouse, ports (Local names: Bargin, Parchin, Barjin, Barajin) are the remains of Lake | 800 m |

---

## Author Comment (AC2)

Appendix 1

Document of historical text

(Siroux, 1949)

**فصل اول**

**نظر اجمالی بر راههای بازرگانی ایران**

احتمالاً" ساکنان سرزمین ایران بسیار زود از حیوانات بارکش برای حمل ونقل کالای خود استفاده نمودهاند . بنابر اطلاعاتی که از دوران پیش از تاریخ ایران در دست داریم ، اینطور بنظر میرسد که پس از اینکه دریاچهٔ مرکزی ایران بتدریج خشک شد هوای آن نواحی هنوز آنقدر رطوبت داشت که مراتع وسیع اطراف خود را حفظ نماید و در نتیجه زندگی چوپانی فعالی در این نواحی بوجود آمد . احتمالاً" در این زمان مبادلات بازرگانی و درنتیجهٔ تماس های اتفاقی قبیله ها باوساطت دوره گردها انجام میگرفت (۱) .

معهذا حتی پیش از انقلاب دوم عدهای از این مردم اینطور استنباط کرده بودند که حمل ونقل کالا و کارهای بازرگانی ممکن است عواید خوبی داشته باشد .

هنگامی که قسمت مهمی از این دریاچه ها خشک شد و بقیهٔ آن نیز درحال خشک شدن بود ، در دهانهٔ رودخانه هایی که وارد این دریاچه ها میشدند و در قسمت های پست دره ها اجتماعات کوچکی مستقر گردیدند که قسمتی از سال را بصورت شهرنشینی بسر میبردند و قسمت دیگر را مشغول گله چرانی میشدند و بعضی از آنها" نیز کاملاً" شهرنشین شده بودند . ویرانه هائی که اکنون بصورت تپه های کوچکی در این نواحی دیده میشود از آثار همین اجتماعات اولیهٔ انسان ها در سرزمین ایران است . این تپه ها که بیشتر در کنار جادهها و کوره راهها قرار گرفتهاند نشانهٔ آنست که حتی از زمان های بسیار قدیم خط سیر انسان در این سرزمین تغییر نکرده است .

۱) احمد ابن الحسین الکاتب در تاریخ جدید یزد در اواخر قرن نهم هجری مینویسد : " ...... پساز

چندی یزد گرد ( ۴۴۸ـ۴۵۷ میلادی ) پسر بهرام گور تیسفون را ترک کرده به ناحیهٔ فارس رهسپار شد . وی

دستور دادساختمان های یزد را زودتر تمام کنند . در کنار او سرهنگی بود ( میگویند سه سرهنگ بود )وسومین

آنان میبد نام داشت . هرکدام شهری ساختند ، میبد ، بیدارو اغدار . شهر اخیر شهر گبرها نیز نامیده

میشود . این سه شهر یا دهکده در کنار دریاچهٔ ساوه قرار داشتند این دریاچه از ساوه تا همدان در فاصله

یازده فرسنگ ( ۶۶ کیلومتر ) از یزد قرار داشت . در میان این دریاچه کوه تجن قرار گرفته بود .بندریبنام

ده بر چین در کنار آن بود که امروز ده برگین نامیده میشود . میگویند هنگام تولد پیغمبر اسلام این دریاچه

خشک شد . آتش آتشکده ها نیز خاموش شد و طاق کسری در تیسفون شکست برداشت . ( ترجمهٔ این مطلب

بوسیلهٔدکتربهنام انجام گرفته است . و نیز مراجعه کنید به کتاب های گوبینو و دیولافوآکهقبلا "نام آنهاگذشت ).

و او را پسری بود یزدگرد نام کردند همنام پدر و یزد بهاقطاع او داد و در وجه دایه و دبیرستان او نهاد. تا بر این مدت بیست سال برآمد و شاهزاده یزدگرد چون سروروان بیالید وگل رخسارش در باغ دولت بشکفت. شاه بهرام گور همسر شایسته در کنار او کرد و بعد از مدتی شاهزاده از طیسفون با حرم و خاصگیان متوجه فارس شد و از آنجا به یزد آمد. و چون عمارت یزد نیم کار مانده بود بفرمود که بنایان دیگربار به عمارت مشغول شدند و از همه مملکت خانه کوچ بیاوردند و در یزد مقیم کردند.

[٣٤]

و دو سرهنگ همراه داشت: یکی را نام بیدار و دیگری را نام عقدار. وگویند سه سرهنگ بودند، سوم را نام میبدار. و ایشان هر یکی دهی بساختند. بیدار «بیده» بساخت و عقدار «عقدا» بساخت که به «ده گبران» مشهورست و میبدار «میبد» بساخت. و این هر سه ده بر کنار دریای ساوه بود، و از ساوه تا همدان و تا ده فرسنگی یزد این دریا بود و چند کوه در میان این دریا بود و بندر او «بارجین» بود که امروز او را «بارگین» میخوانند.

و از معجزات مولود حضرت پیغمبر آخرالزمان صلیالله علیه وآله وسلم یکی آن بود که در آن شب آب این دریا خشک شد و آتش تمام آتشکدهها بمرد و طاق ایوان کسری بشکست، چنانکه گفتهاند:

شعر

آنشب که ز مادر او جدا شد     عالم همه از بلا رها شد

هم آتش تیز فارس مرده     هم آب سیاه ساوه برده

(Eghtedari, 2022)

**دریاچهٔ ساوه**

بین علیآباد و دریاچهٔ مزبور دو فرسنگ شود و دریاچهٔ ساوه در شب ولادت حضرت رسالتمآب محمد مصطفی صلیالله علیه و آله و سلم دفعتاً خشکیده و در آن فرخنده شب چند حادثه در عــالم اتفاق افتاده چنانچه «صباحی بیدگلی» فرماید:

| نم رود سماوه، خشکی دریاچهٔ ساوه | خمود نار آتشخانه، کسر گنبد کسری |
|---|---|

رود سماوه دریای نجف اشرفاست و دیگریفوراندردریاچهٔ ساوه است و دریای نجف در چند سال قبل خشکیده و حال در مــوضع آن ابنیه و باغات زیاد ساختهاند و دریاچهٔ ساوه در سال هزار و سیصد و چهار آب آورده و حال میگویند ممکن است کشتی بــر آن عبور و مرور کند و ماهیهای بسیار بزرگ در آن دیده شوند. تا ببینیم باز از خیام غیب بعرصهٔ شهود که قدم گذارده است.

تسعیر ارزاق در علیآباد نان یکمن تبریز یکهزار و پنیر شش هزار و جو هشتصد دینار و کاه سیصد دینار و هیزم هفتصد دینار و خربزه پانصد دینار و انگور یکهزار و ماست یکهزار و دویست دینار و مرغ یک قطعه دو هزار و پنج شاهی بود.

(Persia, 1888)

**On the New Lake between Ḳom and Ṭeherân.**
**By H.M. The Shâh of Persia.**

(From the Ṭeherân gazette 'Irân,' Nos. 655 and 656, 10th and 19th May, 1888 : translated and annotated by General A. Houtum-Schindler.)

Map, p. 676.

THE lake which has appeared between Ṭeherân and Ḳom is the lake of Sâvah, of which mention is made in history, and which dried up about 1357 years ago, on the day the prophet—may the blessings of God be upon him and his posterity !—was born. It reappeared about six years ago.*

> \* De Sacy fixed the date of the prophet Muḥammed's birth as the 20th April, A.D. 571, but added, " En vain chercherait-on à déterminer l'époque de la naissance de Mahomet d'une manière qui ne laissât subsister aucune incertitude." Sprenger follows him, but considers the date only as a conventional one, generally accepted during the first half of the first century of the Hijrah. Muir accepts, as an approximation, the date

Going toward Kom from Manzarieh one can plainly see that the whole five farsakhs of road one passes over from Manzarieh to the, fields one farsakh from Kom were formerly covered by a lake, *This part of the country was known in olden times as "chal i derya" (the bed of the sea), and is so called now. Several sea-shells were picked up there, and probably in future ages this district will once again become covered by a lake, as it formerly was*†. One farsakh from Manzarieh the road to Kom passes the Savah river, which flows from west to east, bends to the north below Pul i dellak, and then flows into the lake we have described. Here where the road passes the river a noble bridge has been built at the expense of Government. The Amin i Sultan

(Gabriel, 1939)

در نقاط مختلف کرانه کویر تراسهایی به چشم می خورد که نشانهٔ خط ساحلی یک دریاچهٔ قدیمی است. این تراسها درحین بزرگترین دورهٔ گسترش کفهٔ کویر، که اکنون بکلی خشک شده، به وجود آمده است.

نخستین محقق دوران اخیر که با دیدعلمی به کویر پرداخته «اف. آ. بوهزه[30]»، گیاهشناس روسی است. وی در آوریل سال ١٨٤٩ در فصل مناسب سفرش را آغاز کرد و درجهت خط استوا با عبور از «دامغان» و «جندق»، کویر بزرگ را پشت سر گذاشت. در آن زمان به نظر می رسید که غالباً با اسب که خطر کمتری داشت است از باتلاقهای نمک عبور کرده باشند، «بوهزه» نیز در کویر بقایایی از دریاهای محصور و مسدود دوران اولیه را دیده است.

29. Sabkha  30. F. A. Buhse

ما محل اتراق شمارهٔ ۵۲ در «ترود» را با رغبت ترک گفتیم. برنامهٔ بعدی ما قسمتی درکویر و قسمت دیگر درمرزهای آن صورت میگرفت. قسمت باتلاق نمک که اکنون در شرق ما گسترده شده بود، طبق گفتهٔ آشنایان با منطقه، به نظر می رسید که برای انسان و برای حیوان قابل عبور نیست. جریانات متعدد سرازیر شده از کوه که از شمال و شرق به کویر منتهی می شدند، آب خود را همواره به کویر می ریختند. خشک شدن بقایای دریای گذشته دراین جا به کندی صورت میگرفت.

تردیدامیز است. ما سهم بسیار مهمی از شناخت خودمان دربارهٔ کویر را مدیون محقق انگلیسی «اچ. بی. وگان» هستیم. این محقق خستگی ناپذیر کویر، همواره به سوی مناطق خشک پرت افتادهٔ شرق ایران، یعنی نواحی نزدیک «خراسان» کشیده شده است. او در سالهای ١٨٨٨، ١٨٩٠ و ١٨٩١ از راههای مختلف در سفرهای متعدد این مناطق را زیر پا گذاشته و در این میان از موقعیتهای مطلوب برای تعیین مرزهای باتلاق نمک استفاده کرده است. اوچندین بار نیز به سطح کویر بزرگ رفته است و در سفر خود از «خور» به «طبس»، مسیر کرانهٔ جنوبی خلیج شرقی کویر خراسان را از همان راهی که «سی. ام. مک گرگور[33]» در شانزده سال پیش رفته بود، پشت سر گذاشته و به همراه «سی. اف. بیدولف[34]» از مسیله عبور کرده است. کوشش برای تحقیق در «بند گتل» از سمت شمال بینتیجه ماند. اطلاعات مختصر دربارهٔ دو رودخانهٔ کال مورا و «کال لدا[35]» را که از قرار معلوم به یک دریاچهٔ بزرگ می ریزند، مدیون او هستیم. وی از داستانهایی نقل قول میکند که از کویر در آنها به عنوان دریایی با کشتیها، بندرگاهها، برجهای فانوس دریایی و غیره یاد شده است.

(Pirniya and Afsar, 1991)

راههای پیوندی :

در میان استانهای بزرگ پارس خراسان کرمان و شبستان وکوهستان واصفهان بیابانهای پهناوری است که گهگاه درگوشه وکنار آن آبادیهای کوچك بچشم میخورد . اگرچه تا سدههای چهارم وپنجم هجری مانند امروزیکسره خشك وبیآب وگیاه نبوده گاهی دریاچه ساوهگسترش مییافته و با پیوستن بشورابها ودریاچههای نمك وپارگینها از ورامین تا نائین ومیبد کشیده میشده ودرختان تاغ وجفته وگز وبوتههای خار آدور تااندازهای ازبرخاستن ریك وروان شدن آن پیشگیری میکرده است با اینهمه هرگز شهرو آبادی قابل توجهی که بتواند مردمی را در خود گردآورد درمیان دشت برپا نشده تنها بر کران این پهنه چند شهر کوچك چون بندری بر کناره دریای ریك خودنمائی میکرده است .

۱۱۰

برفراز برخی ازمیلها ومنارهها درشب وروزهای مهآلود وطوفانیوتاریك مانند برجهای دریائی آتش نیز میافروختند و آثارآتش درمیلهای اخنجان وراد کان درجای مناسب یعنی میان دوگنبد مؤید این نظر است وبرج نور آباد ممسنی فارس (که آن نیز درکنار شاهراه فارس بوده) بافروغ خود رهگذران را راهنمائی میکرده است وهنوزکه هنوزاست روشنیآن فراموش نشده تاآنجاکه نام خودرا بآبادی نور آباد داده است .

۱٤٤

درجای دیگر درباره میلهای دریائی سخن رفت وبرجهائی راکه بنام خشابه وخشبات معروف بوده (وگمان کردهاند ازخشب تازی بمعنای چوب گرفته شده باینکه هردو واژه گویشهای ایرانی وبمعنای مانوردهنده و گنبد وبنای رخشنده است) . نظیر این برجها در کنار پارگینها (پارچین) ودریاچههای دوونجد ایران نیز هست وازآنجمله برج پارچین ورامین است که روزگاری در کنار پارگینمتصل به دریاچه ساوه برپای شده است .

۱۵۵

(Haghighat, 1962)

**علت بنای اولیه سمنان(۲)**

گویند درحدود دوهزارسال قبل از میلاد مسیح طهمورث دیوبند بر اریکه جهانبانی نشسته وایران را نیز ازهرحیث دایرساخته بودطهمورث در اوایل سلطنت خودبرای رسیدگی بامور کشورعازم سرزمین پارت(خراسان) گردید و در حین مسافرت از محل سمنان که در آن زمان بصورت جنگل و ساحل دریا بوده است دیدن کرد . سمنان در آن زمان جلگهٔ وسیع و خوشمنظرهای بود که نظیر آن نقطه را نقاط مازندران و گیلان حالیه باید دانست در آن روزگار درهشت فرسنگی جنوب سمنان دریای ساوه موج میزد ورودها ازهرسو بطرف دریا روان بودند، سراسرسواحل این دریا پر ازسبزه ودرختان جنگلی بود و از آواز بلبل وهزاردستان ثانی باغ جنان شده بودطهمورث وقتی که باین سرزمین رسید خرم و شادان گشته چند روزی توقف نمود.

اکثر مؤلفین کتب تاریخی و جغرافیائی طهمورث را بانی شهر سمنان دانستهاند . وازطرفی بامراجعه به بخش تشکیل فلات ایران در همین کتاب ثابت میشود که درجای کویرحالیه ایران دریای محصوری بوده است که رفتهرفته آب آن تبخیر وخشک شدهاست. پس بامر طهمورث شهری دراینمکان بنا گردید و نخستین

(Hedin, 1910)

قسمتها بوسیلهٔ دورهای میان رودخانهای با رودهای کوتاهتر و دریاچههای کوچك شده
از هم دیگر جدا شدهاند . همه چیز حاكی از آن است ،که در زمانهای بسیار پیش در
مقایسهٔ با امروز بارشهای زیادتر وپر نتیجهتری وجود داشته است. حتی ارقام تاریخی
وافسانهها و نقل قولهایی که به ما رسیده است با این ادعا میخواند . ظاهراً آخرین
دورهٔ آبهای زیاد وپر نتیجه زمان اسکندر ـ ۳۰۰ سال پیش ازمیلاد ـ واستخری ۹۰۰
سال پس از میلاد ـ را شامل میشود . ظاهراً حد فاصل بین پرآبی بزرگ در زمانهای
بسیار کهن و خشکی عهد جدید حد فاصل بین آخرین دورهٔ رودخانهای و دورهٔ میان
رودخانهای جدید است . هونتینگتون به این سئوال،که آیا دلیل مستقلی که حاكی از
دگرگون شدن آب و هوا در عصر تاریخ باشد وجود دارد ، جواب مثبت میدهد .
چون اسکندر و استخری ثابت میکنند ،که آب وهوای امروز خشكتر است .

خود ایرانیها در جندق این احساس را دارند ، که در ساحل دریا زندگی میـ
کنند . بنا به روایت ، کویر در زمان انوشیروان ، یعنی ۱۳۵۰ سال پیش ، دریاچهٔ
بزرگی بوده است . در سمت غربی این دریاچه رود بزرگی بهنام قاراچای ، که از
همدان وساوه میآمد ، به دریا میپیوست .

جالب توجه است ،که بدانیم دهکدهٔ یونس همان نامی را دارد که یونس پیغمبر و این
پیغمبر بنا به افسانهای که در این ده تعریف میشود در ابن ده از دهان نهنگك افتاده
است وهم چنین دروازهٔ جندق از آوار یك كشتی ساخته شده است که در دریای کویر
رفت وآمد داشته است وحسینان ودیگر روستاهای حاشیهٔ کویر روزگاری بندر بودهاند.

فرورفتگی دریاچهای قدیمی زیبا ومنظمی هم داریم به نام فرورفتگی طبس .
این فرورفتگی در عصر یخبندان دریاچهٔ مهمی را شامل میشده است .این تراس
دوطبقهایکه درراه پرواده ـ همواره درسمت چپ راهـ قرار دارد به وضوح مشهود
است . این تراسها مانند ردیفی از تپههای هم ارتفاع پیوسته به چشم میخورند و
جبههایگرد دارند و به وسیلهٔ برشها و شیارهای فرسایشی ازیک دیگر جدا شدهاند .
من در گودترین قسمت این فرورفتگی دریاچه نمك کوچك و کم عمقی یافتم به نام
«آبكویر» . این دریاچه آخرین بازمانده دریاچهٔ عصر یخبندان است .

در اینکه فرورفتگی طشت دشت لوت هم در دورهای با آب و هوای مرطوب
بستر یك دریاچه بوده است بیشتر به این خاطر تردید نداریم ،که این فرورفتگی در
مرز سیستان قرار دارد و فقط با سیستم کوهستانی پستی از این استان جدا می شود .
عللیکه هامون را به صورت بسیار گودتروگستردهتری در آوردهاند بایستی در دشت
لوت هم کهگودالی بسته است و از کوهستانهای اطراف خود رودخانه های زیادی را
به طرف خود میکشد نقشی داشته بوده باشند . درراه حاشیهٔ شمالی لوت ازبسترهای
زیادی میگذریم ،که وسعت آنها متناسب با بارشهای امروزی این سرزمین نیست و
به همین ترتیب همهٔگودالهای بیرودخانهٔ ایران ،که بعضی ازآنها هنوزهم دریاچهای
موسمی ویا حوضچههای نمك به وجود میآورند، روزگاری بستردریاچههای بزرگ
و کوچکی بودهاند . بدون تردید بیشتر این دریاچهها از فراز بلندیهاییکه فقطکمی
از سطحکویر بلندتر هستند با همدیگر در ارتباط بودهاند .

درعوضتردیدی نیست،که تراس رسوبی کاملاً مشخص طرود یکی ازکرانههای
پیشین دریاچهای بوده است ، که در اینجا قرار داشته است . فقط یك دریاچهٔ
بزرگ ، که برای مدتی طولانی حوزهٔ معین را اشغال کرده است میتواند یك چنین
نشانهٔ مشخص و واضحی از خود برجایگذاشته باشد . بدون شك رسوبهایی از این

Afshar, I., 1978. New History of Yazd. Publications of Farhang-e Iran Zamin, Tehran, Iran (in Persian), 320 pp.

Eghtedari, A., 2022. Sadid Alsaltaneh Travel Book. Sokhan, Tehran, Iran, 739 pp.

Gabriel, A., 1939. Aus den Einsamkeiten Irans. Strecker und Schroder Verlag, Stuttgart, Germany 186 pp.

Haghighat, A., 1962. The History of Semnan. Etelaat Publication, Tehran, Iran (in Persian), 236 pp.

Hedin, S.A., 1910. Overland to India. Macmillan and Company, limited.

Persia, S.o., 1888. On the New Lake between Ḳom and Ṭeherân. Proceedings of the Royal Geographical Society and Monthly Record of Geography, 10(10): 624-632.

Pirniya, K. and Afsar, K., 1991. Road and Robats. National Antiquities Protection Organization of Iran, Tehran, Iran, 220 pp.

Siroux, M., 194٠٩Caravansérails d'Iran et petites constructions routières. l'Institut français d'archéologie orientale.